# Focus or Neglect on Cognitive Impairment Following the History of Multiple Sclerosis

Ugo Nocentini [1,2]

1   Department of Clinical Sciences and Translational Medicine, University of Rome "Tor Vergata", 00133 Rome, Italy; u.nocentini@hsantalucia.it
2   Istituto di Ricovero e Cura a Carattere Scientifico "Santa Lucia" Foundation, 00179 Rome, Italy

**Abstract:** Cognitive disorders are now considered an integral part of the picture of multiple sclerosis. If we trace the history of the accounts of this disease, from the early descriptions by Jean-Martin Charcot, the first to provide systematic characteristics of multiple sclerosis, to present-day accounts, reports of cognitive disturbances have demonstrated an alternating trend. Cognitive disturbances were identified in the beginning, quite clearly for the times. Then, for a long time, they were considered infrequent or attributed to other factors. Finally, since the 1980s, cognitive disturbances have been the subject of increasingly in-depth studies, and are currently assumed to be a very important consequence of multiple sclerosis. In this work, the history of the description of cognitive disorders of multiple sclerosis will be retraced by analyzing the possible reasons for the differences in attention they have received over time. It emerged from the analysis that, as in the case of other pathologies, various factors have influenced how cognitive disorders have been taken into consideration. Some of these factors are inherent to the very nature of the cognitive impairments present in multiple sclerosis; others are linked to historical periods, or to the different ways of approaching the analysis of the phenomena caused by a disease. The reflections made on these topics should, among other things, increase our awareness of how scientific investigation is invariably placed in the historical context in which it is carried out.

**Keywords:** multiple sclerosis; history; cognition; focus; neglect

## 1. Introduction

Multiple sclerosis (MS) is a chronic, inflammatory, demyelinating, and degenerative disease of the central nervous system (CNS). The presence of focal lesions (the so-called plaques), diffuse lesions (axonal degeneration, a diffuse inflammation affecting both the white and gray matter outside the focal lesions [1]), synaptic failures [2], and alterations in the metabolic pathways and energy balance (e.g., mitochondrial resilience) [3] leads to a multifaceted and variable clinical picture which can be different from one patient to another, but also in the same patient at various stages of the disease [4].

Tracing the history of MS, from the first descriptions of the disease to the present day, is not only important from a historical or social viewpoint: the reconstruction of this path allows us to understand, for example, how the cognitive impairments and psycho-emotional disorders which characterize MS are under-recognized or variously interpreted, even today.

The aim of this review is to retrace the history of the identification and definition of cognitive disorders within the clinical picture of MS, and to analyze how much attention these disorders receive. Furthermore, an attempt is made to interpret changes over time, the long phase of apparent oblivion towards cognitive impairments in particular.

The biographies and scientific publications considered in the present paper are those considered most significant or suitable for illustrating the prevailing thought in each era.

The paper is divided into three sections: the "before Charcot" era; the era of Charcot and other contemporary scholars; and from Charcot to the 1980s. There is also the period from 1991 to the present day. Although this final period is more than 30 years long, it must be considered current, and will therefore not be treated in this paper.

## 2. The "Before Charcot" Era

*First Historical Cases of MS*

The examination of the first cases of presumed MS is possible only based on biographical material and would lead to the conclusion that, in these patients, there were no alterations in cognitive function. For some of these cases, MS diagnosis is questionable and has been debated by highly experienced authors (e.g., [5–8]). Other difficulties stand in the way of identifying possible cognitive impairments: e.g., the interference of mysticism; ecstasies; and consumption, in the case of Santa Liduina of Schiedamm [9]; or the great intellect and, therefore, the high-level cognitive reserve in the cases of Heine, Gatty, and Cummings [10,11].

Sir Augustus D'Este is a historical case for which the diagnosis of MS is most likely correct. We can rely on his own account of the disease and its development over a period of 26 years. From this report, no elements of cognitive impairment seem to emerge; however, as we know from recent studies [12], the patient's awareness of their cognitive deficits may not correspond to the real situation. However, no mention of cognitive deficit is present in the illustration of the case of Sir Augustus D'Este, even by contemporary scholars ([5,10,13]).

Among the first cases whose clinical characteristics were described, even if not fully, are those reported by [14]. His pathological findings are credited among the first reported (if not the first) and relate the case of the cook Darges, in which occurrences of pathological laughing and crying are described but normal intelligence is stated.

For the purposes of this paper, the anatomopathological descriptions made by [15] are of no relevance, as they are not accompanied by significant clinical data.

The report of another early MS case by [16] relates the disease of Dr. C.W. Pennock, and once again excludes the presence of cognitive impairment.

In the same year, we have Charcot's description of MS in his lectures.

The absence of the mention of purely cognitive disturbances also characterizes other case descriptions that occurred from the early 1800s until after Charcot's description.

The case of the drummer Bock, described in 1810 by C.J.T. de Meza, with the onset of the disease described in 1789 ref. [17], cannot tell us anything about the problem of cognitive disorders because there was no follow-up.

The illness of Allan Stephenson (a builder and "manager" of lighthouses; the uncle of Robert Louis Stephenson) [11] may have been MS, but there was no mention of cognitive impairment.

A more reliable diagnosis of MS may be advanced for the case of Margaret Gatty (1809–1873), a renowned Victorian-era writer [11]. Her pathology, lasting 25 years, had characteristics compatible with MS diagnosis. In this case, there does not seem to have been any consequences on cognitive abilities; however, we must consider a high cognitive reserve.

The case of Bruce Frederick Cummings (1889–1919) took place after MS was identified as a separate pathological entity. Interest in this case is linked to the fact that Cummings offers another accurate, first-person description of the course of his MS. Moreover, his diagnosis was made while Cummings was alive. Cummings reported the disease-related events in a book, *The Journal of a Disappointed Man*, which was written under the pseudonym W.N.P. Barbellion and published in 1919 [10]. Even in Cummings' case, however, though various psycho-emotional aspects are described, no mention is made of actual cognitive disturbances. Again, his cognitive reserve and young age must be taken into account.

Additionally, no typical cognitive disorder appears in the picture of other cases (such as Margaret Davis of Myddle and William Brown) described before the turning

points represented by the observations of Charcot and reported by T.J. Murray in various publications [11,18–20].

### 3. The Era of Charcot and Contemporary Clinicians

*Not Just Charcot—A Leader but also a Team*

As observed by [19,21], only once a disease has been given a name and a frame does a cultural change take place: first in the field of medicine, and then in the whole society. As far as MS is concerned, this could have happened before Charcot's admirable synthesis.

In fact, in 1824, Charles-Prosper Ollivier d'Angers had already reported a case that could be considered "the first modern clinical description of MS" [19] in his treatise, *Maladies de la moëlle èpinière.* The descriptions by Carswell [15] and the previously mentioned descriptions by Cruveilhier [14], probably dated 1841, followed.

Friedrich Theodor von Frerichs provided an accurate description of the clinical and pathological characteristics of a previously unknown disease, giving it the name "brain sclerosis" (*Hirnskleroses*) [22,23] in 1849. The clinical descriptions of von Frerichs were complemented by the work of Valentiner [24], which confirmed the presence of sclerosis on the anatomopathological level.

Von Frerichs' description did not have the resonance that it perhaps would have deserved, possibly because he did not disseminate or develop his observations. This may have been for professional reasons, as his major commitment was to the field of ophthalmology. Von Frerichs should also be credited with being the first to diagnose brain sclerosis (or MS) during the life of a person affected by the disease.

Considering the topic of the current presentation, von Frerichs' description was the first to report the presence of "impaired intellectual functions" in patients with MS.

For a list of other early descriptions, see Murray [20].

At this point, we arrive at Jean Martin Charcot. The original French-language description of the mental/cognitive state of the MS patients observed by Charcoat follows [25]. For a translation in English, see [11].

> «*Il y a un affaiblissement marqué de la mémoire; les conceptions sont lentes; les facultés intellectuelles et affective émoussées dans leur ensemble. Ce qui paraît dominer chez les malades, c'est une sorte d'indifférence presque stupide à l'égard de toutes choses. Il n'est pas rare de les voir tantôt rire niaisement, sans aucun motif, et tantôt, au contraire, fondre en larmes sans plus de raison.–Il n'est pas rare non plus de voir éclater, au milieu de cet état de dépression mentale, des troubles psychiques qui revêtent l'une ou l'autre des formes classiques de l'aliénation mentale*»

Charcot's role in defining the fundamental characteristics of MS has been emphasized so clearly and frequently that there is no need to reiterate it. Regarding his identification of the presence of cognitive disorders, many the recent reports on this aspect of MS begin with the recognition of what Charcot established, based on clinical evaluation alone, more than 150 years ago [26,27].

It is relevant to underline the role of Charcot's lifelong friend, Edmè Felix Alfred Vulpian (1826–1887) [28,29], in the development of the anatomo-clinical correlation method that led to the separation of MS from other diseases of the central nervous system, particularly from Parkinson's disease [30]. Between 1865 and 1866, Charcot and Vulpian took turns describing various aspects of the "new" pathological entity. According to Bourneville and Guérard [31], Charcot and Vulpian were aware of the descriptions made by Turck, Rokitansky, Frerichs, Valentiner, and others. Regarding Valentiner, Charcot emphasized the lack of a rigorous approach to grouping cases. It can also be said that Vulpian contributed, on par with Charcot, to identifying cognitive disturbances as one of the hallmarks of the disease. As reported by Ross [32] who, in turn, cites Ballet [33], Vulpian seemed to have understood that the mental symptoms were inconstant, irregular, and polymorphic, as were the lesions, which were found in autopsies at the time and are identified today using MRI.

Charcot's description evidently spread rapidly: as soon as 1870, in the United States, Clymer [34] published a long report on the "new" disease. He adhered to the subdivision of

MS into cerebral, spinal, and cerebro-spinal forms, believing that memory and intelligence problems occurred as the disease progressed in the cerebro-spinal cases, up to the loss of the latter. He also affirmed that an intellectual weakening could already be present at an early stage.

However, Clymer's report could have had a scarce diffusion; the uncertain place of mental disorders caused by diseases such as MS, of proven "organic" origin [35] in the American medical world, could also have had a role. Another pioneer of the recognition of MS as a specific disease in the USA was Edward Seguin (1878), who is also credited with the first use of the name "multiple sclerosis" [36,37].

Moreover, as reported by Butler et al. [35], various prominent English clinicians, such as Wilks [38], Gowers [39], and Bury [40], in a period of about 30 years, placed the mental disorders of MS in a secondary position.

In England, in a first-described case [41], Moxon reported "a weakened intellect". In his subsequent descriptions of MS cases [42], mention is also made of possible cognitive deficits.

Once again, however, it is not easy to establish whether these are the aspects that we currently consider cognitive or not, or whether they were the emotional and affective disorders that emerged more sensationally.

Clymer [34,42] is positioned in the conceptual line that considered affective and cognitive disorders together (according to Richardson et al. [5]).

## 4. From Charcot to the 1980s

*The Alternation of Points of View*

At the beginning of the twentieth century, the vision of the exceptional nature of cognitive disorders in MS patients does not seem to change substantially. The reports of MS cases insisted on atypicality or exceptionality or hypothesized the superimposition of other pathological conditions to explain the occurrence of mental symptoms. For example, Cestan and Philippe, as reported by Walusinski [43], considered cognitive disturbances (the impairment of memory and intellectual abilities) to be frequent only in the advanced stages of the disease; rather, they emphasized euphoria and the pseudobulbar affect.

The report by Ross [32] in *The Review of Neurology and Psychiatry* is quite representative of the opinions of most contemporary clinicians regarding "mental" disorders in MS.

In the introduction, Ross reported that the "occasional presence of mental symptoms in disseminated sclerosis is recognized by all authorities", even if there was no agreement on their frequency. Charcot, Raecke, and Seiffer are cited as supporters of a high percentage of occurrence. Ross then illustrated 5 cases identified at the Royal Edinburgh Mental Hospital as the only 5 among the 750 hospitalized. This confirms other reports of the rarity in of the mental symptoms that would have led to hospitalization in an asylum in MS.

Of the five cases described, only one showed a clear and relevant impairment of cognitive functioning, including a working memory deficit (arithmetic calculation). Two had mild cognitive impairment. However, psychiatric disorders prevailed, and this justified the admission to the Mental Hospital.

This report from a small cohort is additionally useful because it re-proposes the inclusion, without any distinction, of psychiatric disorders and cognitive deficits in an indistinct group of mental disorders.

In the discussion, Ross focuses, above all, on aspects such as sclerotic euphoria and on what we currently call pseudobulbar affect. Ross states that other authors had underlined the presence of cognitive impairments.

Following Oppenheim's *Textbook on Nervous Diseases*, it is mentioned that Seiffer and Daunenberger attributed a peculiarity, calling it "polysclerotic dementia", to the cognitive and behavioral impairment found in MS patients. Another interesting quote made by Ross is that Seiffer attributed the mental disorders to the pathological (brain sclerotic) process caused by the disease.

In the period between the end of the 19th century and the First World War, considering also the descriptions of Wilks [38], Bury [40], and Gowers [39,44] as well as that of Ross [26],

the following can be said about cognitive disturbances: (1) they were considered mild and rare or unusual by the majority, or frequent by a minority; (2) some considered cognitive disorders a reaction to the disease; (3) in the advanced forms, the mental deterioration could be severe; and (4) the conjunction between affective–emotional disturbances and cognitive disturbances persists.

In the years following the end of the First World War, greater attention to cognitive impairments in MS seemed to emerge: Jeliffe [45] identified frequent difficulties in formulating thought, attentional processes, and memory; Sachs and Friedman [46] established, using a statistical approach, that 15.6% of a sample of 141 cases had mental changes. As highlighted by Butler et al. [35], the vagueness of the term "mental changes", together with other limitations, did not allow for any generalizations. Brown and Davis [47] believed that cognitive impairments (mental deterioration) may or may not have occurred and were, for the most part, mild; they believed that these impairments would progress to varying degrees from case to case and, even in the advanced stages of the disease, severe deterioration was occasional.

Bohmig [48] examined a large sample of patients (318) with MS and carried out a statistical analysis of their characteristics, finding psychological problems in only 14 subjects.

Two reports published 1926 and 1929, respectively, represented a relevant methodological progress but, paradoxically, they hold two essentially opposite points of view.

Cottrell and Wilson [49] concluded, on the basis of the collected data, that affective disorders had such a significant frequency that they should be considered even more pathognomonic than the symptoms of Charcot's triad. On the contrary, more strictly cognitive disorders should be considered very rare. The study has undoubted merits: it was the fruit of a close cooperation between a neurologist and a psychiatrist, and it was the first systematic examination of a large sample, enrolled according to random criteria, to use a formal questionnaire.

A summary of previous studies reveals that some argued for a high [45,50,51] and others for a low frequency of cognitive disorders [47,48,52,53]. Additionally, there was one study that did not distinguish between cognitive deficits and mental disorders [54].

The study obviously has weaknesses when evaluated with current parameters.

The questionnaire used did not contain questions on cognitive state (memory, attention, etc.) and, as it was based on self-reporting, suffered from the inadequate awareness of some deficits by the patients themselves. The affirmation of the rarity of cognitive disorders (only two cases), apart from being inaccurate (the authors wrote that, in a small number of cases, attention was below normal levels), seems to have been based on a clinical evaluation rather than on collected data, as in the case of psychic and affective alterations.

In any case, Cottrell and Wilson's work had a significant influence over the following decades and was substantially replicated by Sugar and Nadell [55].

The second study is that of Ombredane [56]. It is considered by many to be the forerunner of the methodologically modern evaluation of cognitive and affective status of MS patients. Another advancement was the attempt to exclude, with the methods available at the time, possible diagnostic confusion with neurolue.

Berrios and Quemada [57] highlighted the limits and merits of Ombredane's work on the basis of current diagnostic criteria and with the aid of a statistical approach. On one hand, the limits include the questionable location of some symptoms (fatigue among cognitive disorders and fatalism among affective ones) and the inconsistency of the conclusions with the data as such.

On the other hand, the merits include the identification of a category of variables that correspond to what are now considered cognitive disorders, having brought cognitive disorders out of the grouping with affective-emotional disorders, and to have excluded their reactive nature.

The favor Ombredane's work met in France, and possibly throughout Europe, brings us back to what has already been stated about other studies: hypotheses and data receive a

different reception based on the predisposition of the scientific and social environment in which they are presented.

As mentioned above, the study of Sugar & Nadell [55] represented a verification of the work of Cottrell and Wilson with a very similar conceptual approach, even if the semi-structured interview method was not used.

Notwithstanding the differences in the frequency of the various disorders between Sugar and Nadell and Cottrell and Wilson, the overall picture speaks in favor of a considerable frequency of affective disorders. The differences are an increase in optimism and an increase in pessimism, moving in the opposite direction in the two surveys, and the loss of emotional control, which was much higher in the Cottrell and Wilson study than in the Sugar and Nadel study. The first discrepancy is attributed to the different durations of the illness or different frequencies in the feeling of well-being between the two samples.

A work by Canter [58], which is perhaps not quoted often enough, presented several interesting points. There was a definition of psychological deficit, derived from a direct comparison between a previous performance and the current one. In Canter's study, this comparison was possible for 23 of the 47 enrolled subjects, thanks to the availability of an examination with the same instrument, made years before (the Army General Classification Test-AGCT). In addition, all 47 subjects included in the study were subjected, 6 months apart, to two assessments with the Wechsler–Bellevue Intelligence Scale. For the latter scale, the deterioration index was calculated. This index, and the difference in the scores of the two evaluations with the AGCT, were considered direct measures of the presence of a psychological deficit. The subjects were also evaluated with other instruments, and their results were considered indirect measures of the occurrence of psychological deficits. In addition to the group of MS patients, a group of 38 healthy control subjects underwent evaluations.

Based on the results, some aspects were considered which would receive attention many years later: that (a) psychological (cognitive) deficits tend to progress over time; (b) the time interval required to highlight the aforementioned progression is years, while for shorter intervals (e.g., 6 months), there is substantial stability in cognitive performance; (c) deficits can be brought to light with the use of various tools; (d) statistical analyses and, therefore, the use of group studies (also considering a control group), allows the acquisition of more informative data than the evaluation of individual cases; and (e) on the basis of the data obtained in the sub-tests of the batteries used, the possibility that there are deficits in specific aspects of cognitive functioning is taken into consideration; some of these features correspond to deficits currently considered to be prevalent in people with MS.

Canter's study certainly had limitations. These include, first, the use of tools that have been replaced by others considered more specific in relation to the potential deficits of MS patients. Another limitation is that all patients, except one, were male (as much more recent studies have shown, MS male patients have a higher risk of cognitive impairment) [59].

The work of Jambor [60] can be considered a further advance. It was published in the same issue of the *British Journal of Psychiatry*, which also contained Surridge's paper [61] and was dedicated to the evaluation of psycho-affective disorders. This work was a further index of the awareness that cognitive and affective–emotional disorders could and should be separated. The aims of Jambor's study were to assess whether there were impairments in intellectual efficiency in MS patients, to what extent these impairments existed, and with what pattern they presented (the latter aspect was, perhaps, proposed for the first time in these terms).

A series of tools were used. These were divided by cognitive areas, with coverage of the main ones. The evaluation of signs indicative of organic involvement was also proposed.

In addition to MS patients (divided between patients without (77) and with (26) mood disturbance, for a total of 103 patients), three control groups were examined (79 healthy subjects, 35 patients with psychiatric disorders (mood disorders), and 37 patients with muscular dystrophy). These choices were justified by the impossibility of identifying a single group that would allow for the control of the organic and functional aspects of MS and those that could affect cognitive performance.

The statistical approach appeared adequate to the objectives of the study.

The results of the study supported the hypothesis that MS patients have a significantly lower intellectual efficiency than control subjects, with worse performance in several domains: in particular, memory and non-verbal conceptualization and, to a lesser extent, spatial ability and speech functions. It is surprising that MS patients without depression have a greater impairment compared to those with depression. The author, also based on the performance of psychiatric patients, supported the independence of cognitive deficits from mood disorders. A comparison with patients with muscular dystrophy confirmed that cognitive deficits are due to central nervous system damage. It emerged that the impairment affects various and different cognitive areas. The attribution of memory deficits to the prevalent localization of focal lesions in the periventricular regions is interesting. The author pointed out the evidence of the differentiation between verbal and non-verbal conceptualization as a striking result.

More recent studies (see [26], for a summary) have not confirmed some of the findings obtained by Jambor (e.g., the absence of influence of mood disorders on cognitive performance) and the relative conclusions reached by the author; other results have been confirmed and expanded. The merit remains of having conducted a study with specific purposes, of having examined a sample of significant size with a methodology ahead of its time, and of having considered the need for comparable control groups for one or another aspect of MS.

Eighteen years elapsed between Canter's and Jambor's papers. Afterwards, in the 1970s, the frequency of studies specifically investigating cognitive disorders in MS slowly grew.

Only six papers can be found by searching PubMed for publications whose title and/or abstract contains the terms "multiple sclerosis" and "cognitive," in the interval of 1970, January/1980, December. Three of these papers can be considered specific. Another paper can be found using the words "neuropsychological" and "multiple sclerosis".

In order of time, Matthews et al. [62] compared the performance of MS patients on cognitive, sensory–motor tasks and personality tests with the performance of patients with other neurological conditions. The results showed that MS patients differ from those with other neurological pathologies only in tests of motor speed, steadiness, and fine coordination; for cognitive performances, deficits in non-verbal, conceptual abstraction, attention, and speech-perception tasks were identified, but no significant differences were found between the MS patients and neurological patients.

Reitan et al. [63] investigated the respective roles of motor and cognitive deficits in 30 individuals with MS, noting the prevalence of motor versus cognitive impairments.

Beatty and Gange [64] highlighted the presence of memory deficits and put forward hypotheses, both on the most affected stage of the memorization process and on the relationship between cognitive deficits and motor deficits.

Peyser et al. [65] aimed to establish whether the evaluation of cognitive functions using specific tests could have a diagnostic value, and to establish the independence of cognitive deficits from physical impairment, the degree of neurological involvement, and depression; the clinical evaluation proved decidedly imprecise. One peculiar aspect of this study is the hypothesis that cognitive deficits are the consequence of focal lesions of the subcortical white matter.

These works, therefore, begin to pose specific questions. The limited number of subjects examined does not allow us to establish the frequency of impairments and advises caution in generalizing the conclusions.

Using the same search strategy for the period 1981–1990, 56 papers are found. Of these, 15 articles were published in 1990 and 1989, while the other 26 were published between 1984 and 1988. The end of the decade seems to have been characterized by a significant increase in interest in the cognitive function of people with MS. Moreover, as previously mentioned, in 1991, we entered the current situation for research on MS and cognitive disorders with the studies of Rao et al. [66].

The most significant works of the 1981–1990 period are: a study by Rao et al. [67], which evaluated memory impairment in patients with chronic–progressive forms of MS; a review, also by Rao [68], on the neuropsychology of MS; the studies by Rao and Hammeke [69], Kaplan [70], Peyser [71], and Medaer et al. [72], who examined, from various viewpoints, the problems of assessing cognitive functioning in patients with MS; the study by Heaton et al. [73], which compared the performance of patients with relapsing–remitting courses and progressive courses, identifying a significantly greater impairment in the latter group; the studies of Lyon-Caen et al. [74] and van der Burg et al. [75], who evaluated cognitive functioning in patients with recent onset of MS or with clinically isolated syndromes: these studies showed that some cognitive impairments may occur already in the early stages of the disease; the study by Rao et al. [76], focusing on the nature of memory impairment, which suggested a prevalent impairment in accessing the information stored in memory as opposed to encoding difficulties and limitations in storage capacity. This hypothesis was questioned by subsequent research, which seems to support the hypothesis of encoding difficulties [77]; early work attempting to correlate cognitive impairments with CT or MRI lesion locations [78–83]; yet another study by Rao et al. [84] that evaluated information-processing speed; the papers of Beatty et al. [85–88] and of Jennekens-Schinkel et al. [89–94]; and the paper by Minden et al. [95] that again investigated the nature of memory impairments and their relationship with demographic and clinical variables.

Finally, there were neuropsychological research guidelines by Peyser et al. [96].

Two aspects can be underlined based on the works of the 1980s: the refinement of the methodology due to an ever-greater reference to the dictates of clinical and experimental neuropsychology, with the consequent proposal of increasingly targeted questions; and the arrival of neuroimaging, with investigative possibilities that were unimaginable only a few years earlier.

It is beyond the stated scope of this paper to address the findings of the past 30 years in the field of cognitive impairment and MS, because numerous prestigious reviews are available on the knowledge accumulated from the last decade of the last century onwards to which we refer the interested readers [97–100]

## 5. Discussion

As has been pointed out by distinguished authors [19,21], only after a disease has been identified and given a name do changes occur, allowing for advances in the knowledge of the disease itself and leading to a different attitude in society. Such changes usually occur first in the medical field and then in the social field. Regarding MS, its classification and denomination took place between the first and second half of the nineteenth century, a century characterized by a great "scientific" fervor.

Given the objectives of the present work, this is not the place to examine whether MS appeared at the beginning of the 19th century, or if it has affected human beings since remote times but was only identified in the nineteenth century due to new methods and paradigms of investigation. Regarding the disease's change of name, from the term proposed by Vulpian and Charcot ("sclerose en plaque" or "sclerose en plaque disseminee") to the now almost universally accepted "multiple sclerosis", see [101].

Charcot had already correctly inserted mental disorders (and, among these, cognitive impairments) in the initial descriptions of MS; why did these disorders not receive the same attention as other consequences of MS for a long time? Among other things, others had also identified the presence of cognitive impairments (e.g., von Frerichs [22] and Valentiner [24]; Clymer [34]; and Moxon [42]).

Historical analysis seems to provide us with possible answers.

As previously reported, various scholars identified a previously unidentified pathological entity before Charcot and Vulpian; in particular, Von Frerichs and Valentiner had given it a name (brain sclerosis–Hirnskleroses) and had described its important characteristics; the first diagnosis "in life" is also attributed to von Frerichs.

However, as in other cases (e.g., Alzheimer and Perusini, regarding Alzheimer's disease–[102]; and Dax and Brocà, regarding the localization of lesions that cause aphasia–[103]), the scientific community generally attributed the identification of a pathological condition to those who inserted the individual aspects into a coherent framework or followed a conceptual or methodological setting established a priori.

In the case of MS, the declared application of the anatomo-clinical correlation method by Charcot and Vulpian played a decisive role. Furthermore, Charcot investigated MS for many years, (first in collaboration with Vulpian, and then as the leader of a group of enthusiastic disciples) and disseminated the results of his observations as no one else had previously done, thereby fueling his own fame and using it as a guarantee of the correctness of his ideas.

The relevant results of the scientific investigation, beyond an initial intuition, arose from the coordinated work of a team that could devote itself to a research line for a sufficiently long time. At the Salpetrière, this team was formed and remained active for many years on the topics in which Charcot and Vulpian were initially interested. The same, at least with respect to MS, does not seem to have happened in Germany, England, or the USA.

Therefore, why have mental disorders (and, among these, cognitive ones) not had the same fate, at least in France, as Charcot's triad, for example?

In fact, Charcot himself did not seem to consider mental disorders to be a fundamental characteristic of MS that would fall within the archetype of MS and sustain the diagnosis [5]. Furthermore, even Charcot did not seem to separate cognitive disorders from psycho-affective ones.

Moving on to the early 1920s, the difficulty in reaching shared conclusions is probably related to the fact that the data were derived from the study of a single case or small case series: this prevented the control of the inherent variability of differences from one patient to another [5].

A possible explanation of the neglect into which cognitive disorders in MS fell lies in the separation between neurology and psychiatry that occurred between the end of the 19th and the beginning of the 20th century. This explanation is most likely valid for North America (see Butler et al. [35]). Possible reasons for this separation include the supposed dichotomy between body and mind, the spread of the psychoanalytic approach, and the relative lesser importance, as a discipline, of neurology when compared to psychiatry. All these factors may have led neurologists, who cared for people with MS, to scotomize the symptoms of the neuropsychiatric sphere (psycho-emotional and cognitive).

A different situation occurred in Europe. The reasons for the difference may have included the greater influence, for cultural and geographical reasons, of the work of Charcot and other early authors; the accumulating knowledge regarding cognitive disturbances, resulting from other pathologies of the nervous system (e.g., aphasia, agnosia, and apraxia); and the greater prestige of neurology in Europe compared to North America.

It is also necessary to keep in mind that cognitive disorders, rather than motor, sensory, or autonomic disorders, do not appear clearly on even the most careful observation, and are not clearly perceived by the patient themselves. The reduced perception of cognitive disorders by patients and, to a lesser extent, by their family members, may be due to a denial for disorders that create more fear than "somatic" disorders. The disposition of individual patients towards cognitive disorders was also reflected in the patients' associations, which were formed in the aftermath of World War II [11].

Additionally, in most people with MS, cognitive impairments do not reach a significant degree of severity until the advanced stages of the disease. In the past, a possible yardstick of the seriousness of mental disorders was offered by being interned in an asylum or in a psychiatric institution. If we consider the numbers reported by Ross [32], we must believe that, in many MS patients, mental disorders did not have consequences that led to internment. Furthermore, the living conditions and the average level of health care, until the 1940s, could have favored the death of MS patients before they showed cognitive

alterations of such a degree as to render family management impossible: this management relied on a network certainly wider than that of the present era. Another social aspect that deserves further study for its possible influences on attention to cognitive disorders is the prevalence of MS in females.

The much-reduced consideration for cognitive disturbances was also reflected in Kurtzke's work and his DSS [101,104]; even in the subsequent development of EDSS, the "sphere of mental disorders" was not given a great deal of consideration [105].

A significant increase in the knowledge of a nosological entity arises when one or more paradigm shifts occur: these changes are often linked to the introduction of new methodologies. Regarding MS, the identification of the disease by Carswell, Cruveilhier, Charcot et al. has greatly benefited from the availability of the microscope and histological methodologies. Regarding cognitive disturbances, the methodological innovations are represented by the passage to studies on numerous samples of subjects but above all the introduction of standardized assessment tools. Indeed, the evaluation of groups of people with MS had also been carried out in the first decades of the 1900s (e.g.,: Bohmig [48]; Cottrell and Wilson [49]; Ombredane [56]; Sugar & Nadell [55]). These studies, examined from a current viewpoint, have important methodological limitations which have not made it possible to highlight the cognitive deficits' real extent.

Even for scholars who paid attention to cognitive disorders, a significant obstacle to their identification resided in the possibility of objectification: motor disorders are "self-evident", especially those of walking or balance. The availability of reliable and standardized tools for the measurement of cognitive performance came only with the establishment of Neuropsychology: this corresponds to what we can consider the modern era of research on cognitive disorders in MS.

Another big change brought about by the introduction of neuropsychological methods consisted in the passage from quantitative to qualitative data: the typology of cognitive deficits presented by people with MS had been roughly identified by the first scholars (e.g., Charcot: memory and slowness of concept elaboration). With the studies from 1969 onwards, the characteristics of the memory and attention deficits, and also of impairment of other cognitive domains, have been specified.

The last, for now, decisive step for the investigation of cognitive functions in MS was the advent of modern neuroradiology, in particular MRI. The increasingly sophisticated data derived from MRI techniques, especially unconventional ones, have widened the understanding of the anatomo-clinical correlations of cognitive disorders in MS.

Thus, the method followed by Charcot and Vulpian would still seem valid.

The lack of the overall picture of the locations of the lesions and damages of MS, now provided by the MRI, did not allow scientists, even until the mid-1980s, to hypothesize deficits based on knowledge of the anatomical locations.

## 6. Conclusions

The complete understanding of a pathological entity requires time measurable in centuries rather than decades.

The history of the identification of MS as a separate nosological entity confirms that knowledge of a disease goes through phases in which progress occurs with different speeds. Since knowledge relating to a disease accumulates over many years, the consideration given to the so-called details can also vary according to the change in scientific theories, social orientations and political opinions.

The cognitive impairments caused by MS were identified in early descriptions but have since been seemingly overlooked. In the present work, the most representative studies of both the focus and the low regard for these disorders have been considered. Based on this historical path, an attempt has been made to present the possible origins of the two different attitudes.

If one wanted to attribute an error of assessment to some scholars of the past, this would be the thought that cognitive (or mental, if you prefer) disorders were a mere detail.

Studies conducted over the last 30 years have demonstrated the considerable importance of these disturbances.

However, admiration and respect must prevail for all those who, with a technological endowment not even comparable to the current one but with great observational skills, have made it possible to better understand the characteristics of a complex disease such as MS.

Although important aspects of cognitive functioning in MS patients remain to be clarified, the risk of negligence seems to have been overcome in the case of MS and associated cognitive disorders.

It remains a teaching that could be useful in the study of other pathologies.

**Funding:** This research received no external funding.

**Institutional Review Board Statement:** Not applicable.

**Informed Consent Statement:** Not applicable.

**Data Availability Statement:** Not applicable.

**Acknowledgments:** Thanks to Elisa Porcu, for linguistic proofreading of the manuscript.

**Conflicts of Interest:** The author declares the absence of potential conflicts of interest regarding this work.

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
