# Peer review of "Focus or Neglect on Cognitive Impairment Following the History of Multiple Sclerosis"

_neurosci, doi:10.3390/neurosci4010008_

Round 1
Reviewer 1 Report
Interesting attempt in a poorly studied area, but much work ahead
1. The paper is very wordy and lengthy without reason, please shorten it by around 40%, trying to make your points more simply and clearly.
2. It does not appear well in the paper that the usual cognitive issues in MS are mainly executive and attentional changes, with mental fatigue.
3. The issue of the frequency of neurological dysfunction which does not fit into any anatomical-physiological counterpart, and are thus labeled "functional" (or even "psychiatric") is largely overlooked, although it is a point of major importance when one addresses neurpsychological issues in MS.
Author Response
Responses to the Reviewer 1
Thanks to the reviewer for her/his comments.
1.The paper is very wordy and lengthy without reason, please shorten it by around 40%, trying to make your points more simply and clearly.
It is agreed with the reviewer that the paper has a considerable length. An attempt has therefore been made to reduce it wherever possible. Other parts of the text have not been reduced to provide, in particular, a detailed description of some papers that have not received adequate attention in the past; some of these, among other things, are not easy to find. Also in the discussion, some parts have been abbreviated but others have remained substantially unchanged to plainfully present some fundamental points. The modified parts are shown in red. The eliminated parts are identifiable by comparison with the previous version.
2. It does not appear well in the paper that the usual cognitive issues in MS are mainly executive and attentional changes, with mental fatigue.
The paper is a historical excursus and presents the data and conclusions of research carried out when some distinctions regarding cognitive functions had not yet been formulated. Today, we know that the most frequent cognitive impairments in people with MS affect attention, information processing speed, memory, executive functions and more. But the purpose of the review is not to provide an account of what has been identified in the last 30 years but in the 120 or so years that preceded 1991.
3. The issue of the frequency of neurological dysfunction which does not fit into any anatomical-physiological counterpart, and are thus labeled "functional" (or even "psychiatric") is largely overlooked, although it is a point of major importance when one addresses neurpsychological issues in MS.
The purpose of the review is not the distinction between "organic" and "functional" disorders; a reflection of this distinction emerges, however, in the continuous reference to the distinction between cognitive disorders and psychiatric disorders which was lacking for a long period following the identification of MS as a separate nosological entity.
Reviewer 2 Report
I would like to thank the opportunity to review the manuscript entitled "Focus or neglect on cognitive impairment following the history of multiple sclerosis". The manuscript is fine and brings important topics for discussion to a general audience, especially for neurologists. Some points should be re-evaluated by the authors at this stage:
1. There is frequently the scarce use of references in several paragraphs. For example, in the Introduction, in the first paragraph of the manuscript, despite a detailed description of several aspects regarding Multiple Sclerosis and its characteristics, there is no mention to references.
2. Lines 41 to 43: The description presented in this paragraph is quite confusing, probably due to misleading aspects of grammar use. I recommend the author to perform a detailed review of language aspects related to this manuscript.
3. I think it is not necessary to mention the content described between lines 43-47.
4. It is certainly somewhat frustrating that there is no discussion of current knowledge and research after the 90's, even if it does not represent the aim of the manuscript.
5. Cognitive disturbances are frequently under recognized in MS, even in the current practice. Detailed neuropsychiatric evaluation by specific batteries enables the recognition of subtle changes or subclinical involvement. At the time of the first descriptions, cognitive evaluation by specific tests was not available. Does the author think that this context probably results in underestimated view of this characteristic at this time?
6. It would be better if the authors divide and organize the structure of the manuscript to make comprehension of the content easier for the reader (lines 260-262).
7. At several points, the author divides topic which should be different subsection by using double spaces or spacing paragraphs. Although this represents mainly an editorial topic and style, it would be better in several subsections to divide it in new sections and topics.
8. The conclusion section is quite confusing as it should summarize or analyze the main topics previously discussed in the manuscript and gives the take-home messages for readers. However, it seems that the conclusions represent in the manuscript a new extended topic from the Discussion. I suggest minor changes to improve this section.
Author Response
Responses to Reviewer 2.
Thanks to the Reviewer for her/his comments.
- There is frequently the scarce use of references in several paragraphs. For example, in the Introduction, in the first paragraph of the manuscript, despite a detailed description of several aspects regarding Multiple Sclerosis and its characteristics, there is no mention to references.
The reviewer is correct in pointing out the usefulness of additional references. Therefore, these references have been introduced in several parts of the text.
2. Lines 41 to 43: The description presented in this paragraph is quite confusing, probably due to misleading aspects of grammar use. I recommend the author to perform a detailed review of language aspects related to this manuscript.
A detailed review of the linguistic aspects of the paper has been carried out. In particular, the sentence reported by the reviewer has been substantially modified.
3. I think it is not necessary to mention the content described between lines 43-47.
The contents of lines 43 to 47 have been deleted.
4. It is certainly somewhat frustrating that there is no discussion of current knowledge and research after the 90's, even if it does not represent the aim of the manuscript.
While understanding the reviewer's comment, both based on the purpose of the review (which we have tried to define more clearly) and in order not to increase the already substantial length of the paper, reference has been made to some recent reviews in which it is possible find excellent information on the most recent acquisitions.
5. Cognitive disturbances are frequently under recognized in MS, even in the current practice. Detailed neuropsychiatric evaluation by specific batteries enables the recognition of subtle changes or subclinical involvement. At the time of the first descriptions, cognitive evaluation by specific tests was not available. Does the author think that this context probably results in underestimated view of this characteristic at this time?
The point made was stressed in the discussion and provides an affirmative answer to the reviewer's question.
6. It would be better if the authors divide and organize the structure of the manuscript to make comprehension of the content easier for the reader (lines 260-262).
An attempt has been made to make the structure of the manuscript more suitable for readers' understanding. However, given the topics covered and the data reported, it is not possible to follow the classic scheme of the report of an experimental work.
7. At several points, the author divides topic which should be different subsection by using double spaces or spacing paragraphs. Although this represents mainly an editorial topic and style, it would be better in several subsections to divide it in new sections and topics.
The reviewer is right. The spacings of the previous version had been inserted as an aid for identifying the passage from the analysis of one paper to that of another but it was advisable to eliminate them in the version to be submitted.
8. The conclusion section is quite confusing as it should summarize or analyze the main topics previously discussed in the manuscript and gives the take-home messages for readers. However, it seems that the conclusions represent in the manuscript a new extended topic from the Discussion. I suggest minor changes to improve this section.
The section of the conclusions has been almost completely modified, eliminating references to a building metaphor of little use to the reader and a possible source of confusion.
The modified parts are shown in red. The eliminated parts are identifiable by comparison with the previous version.
Round 2
Reviewer 1 Report
interesting review well revised
Reviewer 2 Report
The author has addressed all the points presented by the reviewer and it improved the quality of the manuscript in several ways. References were properly selected and added to the text in paragraphs which had any previous references.